# Behavioural effects of methylphenidate in the spontaneously hypertensive rat model of attention-deficit/hyperactivity disorder: a systematic review and meta-analysis protocol

Douglas Teixeira Leffa,[1,2] Alana Castro Panzenhagen,[3,4] Diego Luiz Rovaris,[3,4,5] Claiton Henrique Dotto Bau,[3,4,5] Luis Augusto Rohde,[4,5,6] Eugenio Horacio Grevet,[4,5] Gabriel Natan Pires[7,8]

For numbered affiliations see end of article.

**Correspondence to**
Douglas Teixeira Leffa;
douglasleffa@hotmail.com

## ABSTRACT

**Introduction** Attention-deficit/hyperactivity disorder (ADHD) is a prevalent condition related to several negative outcomes, and its pathophysiology is still poorly understood. The spontaneously hypertensive rats (SHRs) are the most commonly used animal model of ADHD. However, its validity, and especially its predictive validity, has been questioned. Therefore, the current protocol discloses the background, aims and methods of a systematic review and meta-analysis of studies reporting the behavioural effects of methylphenidate (MPH), the most commonly prescribed treatment for ADHD, in the SHR.

**Search strategy** Studies will be identified through a literature search using three different electronic databases: Medline, Embase and Web of Science. There will be no language restrictions. All studies that administered MPH to SHR and evaluated locomotion, attention, impulsivity or memory will be included.

**Screening and annotation** Studies will be prescreened based on title and abstract, and a full-text review will be performed if necessary. Screening will be performed by two authors, and any disagreement will be discussed with a third author.

**Data management and reporting** Data extraction will be performed by two independent authors according to a standardised form. Studies will be grouped according to the behavioural outcomes reported, and a meta-analysis will be performed for each group. The influence of predefined covariates on the effects of MPH will be evaluated using meta-regression and sensitivity analyses. Data will be reported following PRISMA guidelines.

## INTRODUCTION

Attention-deficit/hyperactivity disorder (ADHD) is a prevalent neurodevelopmental disorder characterised by impairing levels of hyperactivity, impulsivity and inattention.[1] Observational studies have shown that ADHD is related to several negative outcomes, including, among others, decreased quality of life,[2] increased number of suicide attempts,[3] less socioeconomic status[4] and increased mortality due to accidents.[5]

A strong neurobiological basis has been repeatedly demonstrated in ADHD. Twin studies showed that ADHD has a heritability of about 70%–80%,[6] and significant genome-wide hits have been reported, especially in neurodevelopmental processes that are relevant to ADHD.[7] A recent meta-analysis performed with more than 1500 patients with ADHD found a reduced volume in the nucleus accumbens, amygdala, caudate, hippocampus and putamen in patients with the disorder.[8] In addition, a meta-analysis of functional MRI studies demonstrated reduced activation in distinct cortical region during attention and impulsivity tests.[9] Even though molecular genetics and neuroimaging studies have been improving our understanding of the disorder, the pathophysiology of ADHD is still poorly understood.

Animal models are considered a fundamental tool to unravel the neurobiological factors associated with neuropsychiatric disorders.[10] More importantly, the absence of proper animal models has been proposed as a main factor driving the slow advancement of new treatments for neuropsychiatric disorders.[10] An animal model should present predictive, face and construct validity in order to be considered a proper model of a disease.[11] In ADHD, the spontaneously hypertensive rats (SHRs) are widely considered the most appropriate model.[12 13] Evidence has shown that the SHR presents face validity,[12 13] which corresponds to the extent of similarities between the animal model and the disorder. The evaluation of construct validity, which represents the resemblance between the

| Table 1 | Full search strategies for each database |
| --- | --- |
| Web of science | TS=(SHR OR spontaneous* hypertensive rat*) AND TS=(MPH OR methylphenidate OR ritalin) |
| Medline | ("SHR" OR "spontaneous* hypertensive rat*") AND ("MPH" OR "methylphenidate" OR "ritalin")) |
| Embase | ('shr' OR 'spontaneous* hypertensive rat*') AND ('mph' OR 'methylphenidate' OR 'ritalin') |

aetiological processes in patients and animal model, is still a challenge because the pathophysiology of ADHD is mostly unknown. In addition, there is no consensus on the predictive and construct validity of the SHR as an animal model of ADHD.

Construct validity depends on the understanding of the pathophysiology of the disorder, while predictive validity depends on the similarity between the response to treatment in both humans and animals. Criticism has been demonstrated on the predictive validity of the SHR,[14 15] which might be measured by the ability of the animal model to respond to well-documented treatment for the disorder.[11] In ADHD, the most well-documented treatment is the one performed with stimulant medication.[16] Among the stimulants, methylphenidate (MPH), a dopamine transporter inhibitor, is the most commonly used.[17] Although promoting overall clinical improvement, there is a substantial heterogeneity regarding the response to MPH treatment,[18] and about 30% of patients do not present a clinical response.[19] Therefore, a well-validated animal model is also essential in order to have a proper tool to understand the variability in the response to MPH on a molecular level. Thus, in this study we intend to disclose the protocols for a systematic review and meta-analysis aiming at summarising the behavioural effects of MPH in the SHR. With this study we aim to answer the following question: does MPH treatment improve behavioural deficits presented by the SHR when compared with placebo? In order to achieve this aim, the following objectives were defined: (1) conduct an extensive literature research in order to select all studies evaluating the behavioural effects of MPH in the SHR; (2) extract data; (3) summarise the data using meta-analysis; and (4) explore the influence of predefined variables using meta-regression and sensitivity analysis. Since MPH is the pharmacological treatment most commonly prescribed for patients with ADHD, our results will be important in order to reinforce the predictive validity of the SHR.

## METHODS

Methods are described according to the format proposed by de Vries et al.[20] This protocol has been previously submitted for the Collaborative Approach to Meta-Analysis and Review of Animal Data from Experimental Studies (CAMARADES) Preclinical Systematic Review & Meta-analysis Facility (http://syrf.org.uk/protocols/) with the title 'Behavioral effects of methylphenidate in an animal model of attention-deficit/hyperactivity disorder, the spontaneously hypertensive rats: a systematic review and meta-analysis'.

### Search and study identification

We are going to include studies that administered MPH to SHR and evaluated one of the following behavioural outcomes: locomotion, attention, impulsivity or memory. Studies will be identified through a literature search (Table 1) using three different electronic databases: Medline, Embase and Web of Science. No search filters are going to be used. In addition, we are going to search the reference list of included studies.

### Study selection

The study selection and inclusion is going to be performed with a prescreening based on the title and abstract. If an exclusion criterion is not clearly observed (eg, reviews or editorials, studies in human subjects), a full-text review will be performed. Both will be conducted by two independent authors (DTL and ACP), and any disagreement will be discussed with a third author (EHG). There will be no date or language restrictions. We are going to include experimental studies evaluating the effects of MPH in the SHR in the following behavioural outcomes: locomotion, attention, impulsivity or memory. Only studies that administered MPH to SHR and have a control group will be included in the analysis. The following exclusion criteria will be applied: use of SHR substrains (eg, stroke-prone SHR), MPH administered in brain slices, MPH administered only together with another drug and MPH self-administration. Studies using a cross-over approach, meaning that the same rats will be used as active and placebo groups, will be excluded in order to avoid a carryover effect.[21]

### Study characteristics to be extracted

For the assessment of external validity, the following items will be extracted: study authors, year of publication, experimental groups, sample size, age and sex of animals, route of drug administration, MPH dosage in mg/kg, number of administrations per day, total days of treatment, behavioural test used, and outcome of interest.

### Risk of bias assessment

Risk of bias assessment will be conducted by one review author (ACP) using the SYstematic Review Center for Laboratory animal Experimentation (SYRCLE') risk of bias tool for animal studies.[22] Any doubt in the assessment will be discussed with a second reviewer (DTL). Ten items will be evaluated in the quality assessment: three related to selection bias (sequence generation, baseline characteristic and allocation concealment); two related to performance bias (random housing and blinding); two related to detection bias (random outcome assessment and blinding); one related to attrition bias (incomplete outcome data); one related to reporting bias (selective outcome reporting); and one related to other sources of bias. Baseline characteristics will comprise information

on sex and age of animals. The tenth item will address sources of bias beyond the ones covered by other domains. Each study will be evaluated considering the 10 items, and for each item the study will be classified as presenting low, unclear or high risk of bias. Publication bias will be assessed using funnel plots and the Egger's regression test.

### Collection of outcome data

Continuous outcome data from behavioural tests will be extracted by two independent authors (DTL and ACP), and any disagreement will be discussed with a third author (EHG). Data will be extracted directly from the full-text article. When not reported in enough details, extraction will be done by graph estimation using a digital ruler, as previously described.[23 24] If both methods are not viable, the authors will be contacted. Two attempts will be performed within a 2-week interval. If no response is obtained, the article will be excluded. Each included article will be divided in experiments, defined as any case when a control group is compared with an experimental group. If the same animals were subjected to more than one behavioural experiment from the same category (eg, two distinct locomotion tests), both data will be extracted and the one with the largest effect size will be included in the analysis. In addition, only one outcome will be extracted from each behavioural experiment. Whenever the article reports multiple outcomes from the same behavioural test, the extraction will be performed according to a relevance rank organised by one reviewer. Variables from each behavioural test will be ranked subjectively according to their importance, and the one ranked highest will be extracted. If the same animals were evaluated more than once in the same behavioural test, the last one will be selected for data extraction. If the manuscript divides the results by time, the first time point will be selected for extraction.

### Data analysis
#### Meta-analysis

Studies will be grouped according to the behavioural outcomes reported (locomotion, attention, impulsivity or memory), and a meta-analysis will be performed for each group. A minimum of three experiments will be required in order to conduct the meta-analysis. Pooled effect sizes will be determined with standardised mean differences using Hedge's G method with random effects, allowing the comparison among distinct behavioural tests. The significance of pooled effect sizes will be determined using the Z-test. Since four statistical hypothesis tests will be performed, a Bonferroni correction will be applied in order to control the family-wise type I error rate, and a p value ≤0.0125 will be considered statistically significant. Individual study weights will be obtained using the inverse of the variance. Data will be transformed in order to obtain positive values for decreased impulsive behaviour and for increased attentional or memory performances.

Heterogeneity between studies will be assessed using both the $\chi^2$ and the $I^2$ tests, and a p value ≤0.1 will be considered statistically significant. An $I^2$ value in the order of 25%, 50% and 75% will be considered as low, moderate and high heterogeneity, respectively.[25] Standardised mean differences and heterogeneity values will be obtained using Review Manager (RevMan) V.5.3 (Copenhagen: The Nordic Cochrane Centre, The Cochrane Collaboration, 2014).

### Meta-regression

A meta-regression will be conducted for each behavioural outcome in order to evaluate potential sources of variability among studies. The following covariates will be selected based on biological plausibility and added to a univariate random-effects meta-regression model: age of animals, route of drug administration and MPH total dosage. The age of animals will be categorised into 'adolescent' or 'adult' before being included in the model, since some studies may not report the age in days or weeks. Animals 60 days old or more will be defined as 'adults', while animals 28–60 days old will be defined as 'adolescents'.[26] Route of drug administration will be added as a categorical variable using dummy variables. MPH total dosage will be calculated by multiplying the dosage received in mg/kg by the number of administrations per day and by the total days of treatment, and added as a continuous variable. Covariates associated with the outcome with a p≤0.1 in a univariate analysis will be included in a final multivariate meta-regression model. Since up to four statistical hypothesis tests will be performed in the multivariate model, a Bonferroni correction will be applied in order to control the family-wise type I error rate. Studies with missing values will be excluded from the meta-regression analysis. Meta-regression will be conducted using Stata V.13.0 , as previously described.[27]

### Sensitivity analysis

Sensitivity analyses will be performed in order to evaluate effect size differences related to the main methodological decisions in our manuscript. The following sensitivity analyses will be performed for each behavioural category: (1) the jackknife method, a common procedure used to test the stability of the outcome after excluding one result at a time[28]; (2) including only effect sizes extracted from the same behavioural test, with a minimum of three experiments; (3) including studies with cross-over designs; (4) including only one MPH dosage at a time, with a minimum of three studies using the same dosage; and (5) excluding studies presenting a concerning risk of bias, defined as either a high risk of bias in one category or an unclear risk of bias in seven categories or more.

The objectives of our sensitivity analyses will be to (1) observe if any study skews the overall result; (2) evaluate if the effects of MPH are different for distinct behavioural tests; (3) evaluate possible changes in effect sizes due to a carry-over effect from cross-over studies; (4) observe if the effects of MPH are different for distinct dosages; and (5) evaluate the impact of studies with a high risk of

bias. Sensitivity analyses will not be corrected for multiple comparison, and the results will be interpreted as exploratory. A p value ≤0.05 will be considered statistically significant for these analyses.

## DISCUSSION

The pathophysiology of ADHD is still mostly unknown, and a valid animal model is believed to be essential in order to advance this knowledge. Although the SRH is the most widely used animal model of ADHD, its validity has been questioned by different authors. The poor methodological description, high heterogeneity and high probability of bias observed in preclinical research seem to be a relevant factor for the interpretation of the published literature.[29] In this sense, an overall estimate of the available data obtained using systematic reviews and meta-analyses has become more important over the years. Although they have been already used in the last decades in order to guide clinical practice, systematic reviews and meta-analyses are relatively new in preclinical research.[30]

The predictive validity represents the ability of the animal model to respond to well-documented treatments for the disorder, and is essential for the characterisation of a valid animal model.[11] In this sense, understanding the effects of MPH in the SHR is of great importance since MPH is the pharmacological treatment most commonly used in ADHD. However, we also want to highlight that other pharmacological approaches available for ADHD will not be included in our analysis, thus limiting our conclusions. In the SHR, distinct behavioural responses have been reported after a treatment with MPH. For instance, van den Bergh et al[14] reported no effects of MPH in the animal model of ADHD, while Tamburella et al[31] reported increased locomotion after MPH administration. A deleterious effect of MPH in attentional performance of SHR has also been demonstrated.[32] On the other hand, these results are contradicted by several studies showing decreased locomotion[33–36] and increased attention[33 37–39] after treatment. Even though these divergences might be explained by methodological aspects including MPH dosage, route of drug administration and age of animals, no study has been conducted in order to evaluate this hypothesis.

By conducting this study, we expect to (1) provide an overall estimate of the effect size of MPH on behavioural outcomes when administered to the SHR; (2) provide an estimate of the heterogeneity present in the published literature; and (3) identify covariates potentially related to this heterogeneity using meta-regression and sensitivity analyses. The following are our main hypotheses: (1) MPH treatment will decrease locomotion and impulsivity, and increase attention and memory performance of the SHR; (2) a high heterogeneity will be found; and (3) age of animals, route of drug administration and MPH total dosage are going to be statistically related to the effects.

A detailed description of the methodology to be applied in systematic reviews and meta-analyses has many advantages, including expert external opinion on the methods, prevention of data fishing and the possibility of helping researchers in future applications. Additionally, protocol publication is an opportunity to discuss potential sources of bias that may be present in the future. In this sense, the following factors should be highlighted from our methodology. First, we are going to include only one data point for each behavioural test, which will be selected based on a subjective relevance rank. Although distinct approaches were available for the extraction (eg, the combination of multiple variables into a single data point), we believe that the use of only one variable may ensure increased consistency among results. It is known that the same behavioural test can be performed in order to investigate distinct behavioural parameters. As an example, the open field test can be used to measure locomotion,[40] anxiety-related behaviour[41] or spatial memory,[42] based primarily on which variable is collected. Therefore, even though our approach may present a risk of bias for being based on a subjective evaluation of importance, we believe that the grouping of multiple variables into a single data point would be an even greater threat since it would combine distinct information, thus increasing heterogeneity. Another important aspect is related to our meta-regression model. As previously mentioned, studies with missing values will be excluded from the analysis, which can bias the final results.

To sum up, we believe that the results obtained from this study will be valuable in order to corroborate the predictive validity of SHR as an animal model of ADHD. Additionally, our results will provide data on the methodological strengths of published literature, which may aid the development of new experimental designs. In this way, our study will provide a background for the development of new research and for the advance of knowledge in the area.

**Author affiliations**
[1]Post-Graduate Program in Medicine: Medical Sciences, School of Medicine, Universidade Federal do Rio Grande do Sul, Porto Alegre, Brazil
[2]Laboratory of Pain Pharmacology and Neuromodulation: Preclinical Studies - Pharmacology Department, Institute of Basic Health Sciences, Universidade Federal do Rio Grande do Sul, Porto Alegre, Brazil
[3]Department of Genetics, Universidade Federal do Rio Grande do Sul, Porto Alegre, Brazil
[4]ADHD Outpatient Program, Hospital de Clínicas de Porto Alegre, Porto Alegre, Brazil
[5]Department of Psychiatry, Universidade Federal do Rio Grande do Sul, Porto Alegre, Brazil
[6]National Institute of Developmental Psychiatry for Children and Adolescents, São Paulo, Brazil
[7]Departamento de Psicobiologia, Universidade Federal de São Paulo, São Paulo, Brazil
[8]Department of Physiological Sciences, Santa Casa de São Paulo School of Medical Sciences, São Paulo, Brazil

**Contributors** DTL: Conceptualization, Data curation, Formal analysis, Methodology, Project administration, Supervision, Visualization, Writing – original draft. ACP: Conceptualization, Data curation, Methodology, Writing – review & editing. DLR: Conceptualization, Methodology, Writing – review & editing. CHDB: Conceptualization, Methodology, Writing – review & editing. LAR: Conceptualization, Methodology, Writing – review & editing. EHG: Conceptualization, Methodology, Writing – review & editing. GNP: Conceptualization, Methodology, Writing – review & editing.

**Funding** The authors have not declared a specific grant for this research from any funding agency in the public, commercial or not-for-profit sectors.

**Competing interests** LAR has been a member of the speakers' bureau/advisory board and/or acted as a consultant for Eli Lilly, Janssen-Cilag, Medice, Novartis and Shire in the last 3 years. He receives authorship royalties from Oxford Press and ArtMed. He has also received travel awards from Shire for his participation in the 2015 WFADHD and 2016 AACAP meetings. The ADHD and Juvenile Bipolar Disorder Outpatient Programs chaired by him has received unrestricted educational and research support from the following pharmaceutical companies in the last 3 years: Eli Lilly, Janssen-Cilag, Novartis and Shire. EHG has served on the speakers' bureau and has received travel grants from Shire and Novartis. He has also been on the advisory board and acted as a consultant for Shire.

**Provenance and peer review** Not commissioned, externally peer reviewed.

**Data availability statement** All data relevant to the study are included in the article.

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
