## [Reviewer comments · BMJ Open Science]

ARTICLE DETAILS

TITLE (PROVISIONAL)	Behavioral effects of methylphenidate in an animal model of attention-deficit/hyperactivity disorder, the spontaneously hypertensive rats: a systematic review and meta-analysis protocol
AUTHORS	Douglas Teixeira Leffa (Corresponding Author) Alana Castro Panzenhagen Diego Luiz Rovaris Claiton Henrique Dotto Bau Luis Augusto Rohde Eugenio Horacio Grevet Gabriel Natan Pires

VERSION 1 - REVIEW

REVIEWER 1	Peter Paul Zwetsloot Universitair Medisch Centrum Utrecht
REVIEW RETURNED	31-01-18

GENERAL COMMENTS	.In 'Behavioural effects of methylphenidate in an animal model of ADHD, the spontaneously hypertensive rats: a systematic review and meta-analysis protocol' Texeira Leffa et al. describe their protocol in detail for their systematic review and meta-analysis. This is important to guarantee that the methods used were declared upfront and should be applauded. The protocol is excellently written and includes most important aspects. There are some minor concerns and comments: 1. there is no final search mentioned with all search terms. Please add (in supplementary file) the complete final search that has been inserted in PubMed / Embase etc. Through this, you are absolutely sure that the search used in the primary paper is the same as the one in the protocol.2. (2.2 p7) in '2.2 study selection' the authors state that they will do a full text screening if necessary. I would advice to always do a full text screening to be absolutely sure that your included paper meets al your inclusion criteria (sometimes abstracts can be misleading).3. (2.2) Please state who the authors will be (with initials) who will conduct the search and who will act as third screener/referee.4. (2.2) You exclude SHR substrains (stroke-prone SHR). Is there a reason to exclude these? Are they different from one another? In literature I can find authors saying that both are used as ADHD models. If there is a specific reason to exclude these, please elaborate on this and provide literature on this choice.
--

	5. (2.6.2) Please state how many studies you think you will approximately include, as this officially limits the amount of covariates you can examine. A general rule of thumb is 1 covariate per 10 studies included. If you choose to include more covariates, please state this in the primary paper, as this increases the chance of finding false-positive results. 6. (2.6.2, p10) you will perform a multivariate meta-regression model, in which studies with missing values will be excluded. please be aware that doing a 'complete case analysis' might bias your data and reduce the number of included studies. 7. (2.6.3) you mention sensitivity analysis, based on excluding one result at a time. If you want to do this, to get rid of any outliers I would suggest starting with a 'regular' outlier analysis on your outcome data-points (deviance residuals, Schoenfeldt residuals) and try to exclude those. (NB. This is not a request that certainly needs to be implemented, but just a suggestion.)
--	--

VERSION 1 – AUTHOR RESPONSE

Editor in Chief Comments to Author:

The reviewer has been some excellent suggestions. In addition, although you have clearly stated the aim of the systematic review in the introduction it would be beneficial to include your specific objectives to achieve this aim.

R: Thank you for your comment. Specific objectives were included in the introduction.

I suggest for section 2.5 you make it clear how you will ensure consistency in your relevance ranking of behavioural tests extracted (I assume this means that for each outcome only one behavioural test data point will be included?).

R: Yes, only one data point will be included for each behavioral test. We will rank the variables from each behavioral test subjectively according to their importance, and the relevance rank will be organized by one reviewer. Even though the ranking is organized in a subjectively way, the extraction of all papers will be based on the same rank, thus providing consistency to the method. A similar strategy has been previously used (1, 2).

I also suggest a short description of the pros and cons of this approach should be included in the discussion – i.e. why you have chosen this method rather than to include all data from a behavioural test to nest into a single data point for your outcome.

R: Thank you for your comment. The pros and cons of this approach have been incorporated in the discussion.

Associate Editor Comments to Author:

I agree with the suggestions for minor revisions as made by reviewer 1, and suggest that, if still possible, you try to incorporate these suggestions into your protocol.

Reviewer 1:

In 'Behavioural effects of methylphenidate in an animal model of ADHD, the spontaneously hypertensive rats: a systematic review and meta-analysis protocol' Teixeira Leffa et al. describe their protocol in detail for their systematic review and meta-analysis. This is important to guarantee that the methods used were declared upfront and should be applauded. The protocol is excellently written and includes most important aspects. There are some minor concerns and comments:

1. there is no final search mentioned with all search terms. Please add (in supplementary file) the complete final search that has been inserted in PubMed / Embase etc. Through this, you are absolutely sure that the search used in the primary paper is the same as the one in the protocol.

R: Thank you for your comment. The complete final search has now been added as supplementary material.

2. (2.2 p7) in '2.2 study selection' the authors state that they will do a full text screening if necessary. I would advice to always do a full text screening to be absolutely sure that your included paper meets all your inclusion criteria (sometimes abstracts can be misleading).

R: Thank you for your suggestion. We do agree that abstracts can be misleading. In order to avoid the exclusion of a potentially eligible study and at the same time increase the efficiency of our search strategy, we decided to exclude studies without a full-text review only in those situation that an exclusion criteria is clearly defined (e.g. reviews or editorials, studies in human subjects). The information is now mentioned in "2.2 Study Selection". We would like to highlight that this strategy is advised for both clinical and preclinical meta-analyses (3, 4).

3. (2.2) Please state who the authors will be (with initials) who will conduct the search and who will act as third screener/referee.

R: The authors responsible for the search are now described.

4. (2.2) You exclude SHR substrains (stroke-prone SHR). Is there a reason to exclude these? Are they different from one another? In literature I can find authors saying that both are used as ADHD models. If there is a specific reason to exclude these, please elaborate on this and provide literature on this choice.

R: The stroke-prone SHR are a substrain of SHR created by breeding animals presenting rapid increase in blood pressure at a younger age and severe hypertension, leading to a high incidence of stroke (5). Although few studies have been conducted using the stroke-prone SHR as a model of ADHD, biochemical (6) and genetic (7, 8) differences between the two strains have been reported. Those differences, together with the increase incidence of stroke in stroke-prone SHR, potentially leading to brain damage, prevented us from including the substrain in the analysis.

5. (2.6.2) Please state how many studies you think you will approximately include, as this officially limits the amount of covariates you can examine. A general rule of thumb is 1 covariate per 10 studies included. If you choose to include more covariates, please state this in the primary paper, as this increases the chance of finding false-positive results.

R: Thank you for your comment. Based on our previous experience with basic studies, we believe that at least 30 studies will be included in the final analysis. However, if this is not the case, a statement will be included in the primary paper.

6. (2.6.2, p10) you will perform a multivariate meta-regression model, in which studies with missing values will be excluded. please be aware that doing a 'complete case analysis' might bias your data and reduce the number of included studies.

R: Thank you for your comment. We do agree that this method may bias the final result, and have added a sentence in the discussion section.

7. (2.6.3) you mention sensitivity analysis, based on excluding one result at a time. If you want to do this, to get rid of any outliers I would suggest starting with a 'regular' outlier analysis on your outcome data-points (deviance residuals, Schoenfeldt residuals) and try to exclude those. (NB. This is not a request that certainly needs to be implemented, but just a suggestion.)

R: Thank you for your suggestion. Excluding one result at a time in order to observe how individual studies may affect the final effect size is also called the jackknife method (9), and it is commonly used

in meta-analysis. We have decided to maintain the method, and included more information about it in the text.

References:

1. Pires GN, Bezerra AG, Tufik S, Andersen ML. Effects of experimental sleep deprivation on anxiety-like behavior in animal research: Systematic review and meta-analysis. *Neuroscience and biobehavioral reviews*. 2016;68:575-89.
2. Pires GN, Bezerra AG, Tufik S, Andersen ML. Effects of acute sleep deprivation on state anxiety levels: a systematic review and meta-analysis. *Sleep medicine*. 2016;24:109-18.
3. Higgins JPT GSe. *Cochrane Handbook for Systematic Reviews of Interventions*. 2008.
4. Vesterinen HM, Sena ES, Egan KJ, Hirst TC, Churolov L, Currie GL, et al. Meta-analysis of data from animal studies: a practical guide. *Journal of neuroscience methods*. 2014;221:92-102.
5. Okamoto KaY, Y and Nagaoka, A. Establishment of the stroke-prone spontaneously hypertensive rat (SHRSP). *Circulation Research*. 1974;34:143-53.
6. Ariano MA, Kenny SL. Neurochemical differences in the superior cervical ganglion of the spontaneously hypertensive rat stroke-prone variant. *Brain research*. 1987;415(1):115-21.
7. Yamamoto H, Okuzaki D, Yamanishi K, Xu Y, Watanabe Y, Yoshida M, et al. Genetic analysis of genes causing hypertension and stroke in spontaneously hypertensive rats. *International journal of molecular medicine*. 2013;31(5):1057-65.
8. Churchill PC, Churchill MC, Griffin KA, Picken M, Webb RC, Kurtz TW, et al. Increased genetic susceptibility to renal damage in the stroke-prone spontaneously hypertensive rat. *Kidney international*. 2002;61(5):1794-800.
9. Miller RG. The jackknife-a review. *Biometrika*. 1974;61(1):1-15.

VERSION 2 - Review

REVIEWER 1	Mira van der Naald Universitair Medisch Centrum Utrecht
REVIEW RETURNED	0506/2018

GENERAL COMMENTS	Recommendation Minor Revision Interesting research topic! 7. Note that using a standardized mean difference leads to distortion of the funnel plot. Please see the article by Zwetsloot et al for further details (doi: 10.7554/eLife.24260). 8. I am not in the position to answer this question. 11. Please elaborate how the results will be a measure to predictive validity. Especially describe how the results should be interpreted, taking into account that results of MPH in clinical setting are heterogeneous as well (as stated in the introduction). Furthermore, how will be dealt with heterogeneous internal validity in the analysis? 13. The protocol registered on SYRF mentions 9 items to be evaluated for quality assessment, but in the current manuscript one item is added, please explain why this item is added now. Furthermore, funding and conflict of interests are not addressed.
---

REVIEWER 2	Hendrik Gremmels Universitair Medisch Centrum Utrecht
REVIEW RETURNED	31/05/2018

GENERAL COMMENTS	Recommendation Major Revision The present article is a protocol for a for a systematic review and meta-analysis on the effects of methylphenidate on neurobehavioural characteristics in SHR rats. In the opinion of this reviewer this is an important and relevant subject, that is very suitable for a systematic review. It is commendable that the authors have decided to publish their protocol prior to conducting the review. From the cover letter I understand that the protocol has already undergone a round of review prior to my receiving it. Unfortunately I do not have access to this first review round and will consequently review the article 'de novo'. My apologies to the authors if some issues are addressed doubly.
--

Regarding the general structure of the protocol, the only remark that I have is whether the focus on SHR rats is not overly restrictive. SHR rats are perhaps the most common model for ADHD, but there are other (WKY substrains) that are also proposed models for ADHD. Is the question about predictive validity not better answered by showing that SHR rats show a more 'human-like' response to MPH than other rat strains?

Abstract:

The authors state that the main aim of the article is to examine the predictive validity of SHR rats for ADHD, but they cannot answer this wholly with the present article. They can only show that one documentedly effective treatment in humans, MPH, may also work in the SHR model. This gives only a clue towards predictive validity, as there are other drugs (Amphetamines, Atomoxetine) that work in humans, and perhaps a few that work in SHR and not in humans. The abstract should focus on the actual article, i.e. the effect on MPH and not on speculative implications. For instance, the abstract does not even mention the recorded outcomes of the proposed meta-analysis.

line 5 currently -> current

Introduction

In general the intro could be improved by information why face and construct validity are apparently not a problem in SHR rats. This could be nicely tied in with a bit describing the proposed mechanism of action of MPH.

Par 2, line 4 mega-analysis-> meta-analysis

line 6: spell out fMRI

Par 2, last sentence: fix bad grammar

Par 3, line 2: important->importantly

Page 6 line 6 MPH on a molecular level

Methods

2.2.

It would be better to use the 'conventional' steps in the PRISMA diagram instead of "Step 1-3". I.e. abstract screening, full-text etc. Step 3 should be reversed by specifying inclusion criteria

criteria -> criterion

2.3.

gender -> sex

2.5

The authors have decided to that they will include only one behavioural experiment from the same category and will choose the most relevant or the first. This seems very prone to bias and unclear. The first means the first reported in the article? This may

be a negative bias as authors usually build up towards the more dramatic results in the article. On the other hand relevance decided by the authors of this article may also lead to bias. My suggestion would be to extract all of them and include the one with the largest effect size in the main analysis (effect of MPH). The results of different test variants (e.g. diff locomotion tests) can be presented in separate subgroup analyses (or if need be under sensitivity). Which test 'works' best (and thus has highest predictive validity) is an important question and can refine animal research.

The last two sentences seem vague to me. "If the same animals were evaluated more than once in the same test, the last one will be selected for data extraction. If the manuscript divides the results by time, the first timepoint will be selected..." To my mind these say the exact opposite, please clarify.

line 5: two attempt -> two attempts

2.6

Page 9, last line. Actually the four hypotheses are likely far from independent. Assuming a true underlying construct for ADHD and an effect of MPH thereupon, one expect that relative scores on the four outcome measures are highly correlated and therefore not independent. The Bonferroni correction is conservative and I don't oppose to it, but it may be overkill.

Page 10. Any reason why the authors transform the values for locomotion? Common practice would be to show the negative values and put an additional 'favors MPH'/'favors control' on the axis.

2.6.2

The meta-regression test **are** independent hypotheses, and should be corrected for.

The authors should specify a minimum number of studies (overall, or per arm) before conducting meta-analysis.

Should the test-type (e.g. for locomotion) not be included in the investigation of heterogeneity?